# Intracellular Redox-Balance Involvement in Temozolomide Resistance-Related Molecular Mechanisms in Glioblastoma

**DOI:** 10.3390/cells8111315

**Published:** 2019-10-24

**Authors:** Alessia Lo Dico, Daniela Salvatore, Cristina Martelli, Dario Ronchi, Cecilia Diceglie, Giovanni Lucignani, Luisa Ottobrini

**Affiliations:** 1Department of Pathophysiology and Transplantation, University of Milan, 20090 Segrate (MI), Italy; alessia.lodico@unimi.it (A.L.D.); daniela.salvatore@unimi.it (D.S.); cristina.martelli@unimi.it (C.M.); cecilia.diceglie@unimi.it (C.D.); 2Doctorate School of Molecular and Translational Medicine, University of Milan, 20122 Milan, Italy; 3Neurology Unit, Neuroscience Section, Department of Pathophysiology and Transplantation, Dino Ferrari Centre, IRCCS Foundation Ca’ Granda Ospedale Maggiore Policlinico, University of Milan, 20122 Milan, Italy; dario.ronchi@unimi.it; 4Department of Health Sciences, University of Milan, 20146 Milan, Italy; giovanni.lucignani@unimi.it; 5Molecular Bioimaging and Physiology (IBFM), CNR, 20090 Segrate (MI), Italy

**Keywords:** oxidative stress, chaperone mediated autophagy (CMA), mitochondrial scavenger, drug resistance, reactive oxygen species (ROS), cell motility

## Abstract

Glioblastoma (GBM) is the most common astrocytic-derived brain tumor in adults, characterized by a poor prognosis mainly due to the resistance to the available therapy. The study of mitochondria-derived oxidative stress, and of the biological events that orbit around it, might help in the comprehension of the molecular mechanisms at the base of GBM responsiveness to Temozolomide (TMZ). Sensitive and resistant GBM cells were used to test the role of mitochondrial ROS release in TMZ-resistance. Chaperone-Mediated Autophagy (CMA) activation in relation to reactive oxygen species (ROS) release has been measured by monitoring the expression of specific genes. Treatments with H_2_O_2_ were used to test their potential in reverting resistance. Fluctuations of cytoplasmic ROS levels were accountable for CMA induction and cytotoxic effects observed in TMZ sensitive cells after treatment. On the other hand, in resistant cells, TMZ failed in producing an increase in cytoplasmic ROS levels and CMA activation, preventing GBM cell toxicity. By increasing oxidative stress, CMA activation was recovered, as also cell cytotoxicity, especially in combination with TMZ treatment. Herein, for the first time, it is shown the relation between mitochondrial ROS release, CMA activation and TMZ-responsiveness in GBM.

## 1. Introduction

Malignant Glioblastomas (GBMs) are the most common primary brain tumors and are characterized by a dismal prognosis mainly due to the resistance to conventional therapies. The current therapeutic regimen is based on chemotherapy with Temozolomide (TMZ) combined with radiotherapy [1]. The O6-methylguanine-DNA methyltransferase (MGMT) promoter methylation status is the main biomarker to foresee GBM responsiveness to TMZ, but its predictive value is limited and additional functions of TMZ unrelated to MGMT activity are likely to underlie a further distinction between responders and non-responders due to specific features. For example, TMZ might influence cell viability by increasing reactive oxygen species (ROS) [2,3], or by modulating autophagy [4], apoptosis [5], hypoxia inducible factor (HIF)-1α activity [6,7,8] and epithelial-mesenchymal transition (EMT) [9].

Autophagy plays a critical role in cellular homeostasis: in fact, it is involved both in pro-survival [10] and pro-apoptotic mechanisms [11]. Autophagy also affects the migration and invasion capabilities of tumor cells [12]. Taking into account all these evidences, there are several clues suggesting that autophagy could be involved in the onset and modulation of cell resistance or sensitivity to treatment [13]. Furthermore, different autophagic mechanisms have been described presenting specific drivers, effectors and functional consequences [14].

It is known that TMZ might induce autophagy and that the activation of this pathway is crucial for the susceptibility to the treatment [15,16]. In particular, it has been recently demonstrated that chaperone mediated autophagy (CMA) is the main mechanism by which TMZ treatment decreases HIF-1α activity in sensitive cells, thus improving responsiveness by promoting cell apoptosis [7]. Moreover, the depletion of CMA related genes [17,18] is sufficient to annul the sensitivity to TMZ, demonstrating its important role in the response to this drug.

CMA activity is induced by a transitory release of ROS into the cytoplasm by mitochondria following the endoplasmic reticulum stress. ROS have been described to be regulators of multiple redox-dependent pathways related to growth, differentiation, survival and others [19,20,21]. However, since radicals can impair cell integrity, a redox-homeostasis system exists in cells to detoxify ROS. This enzymatic repertoire includes: Super Oxide Dismutase (SOD), Glutathione Reductase (GR), and Catalase [22]. Irreversible ROS mediated damage might enhance both proteasomal and lysosomal (autophagic) degradation of oxidized proteins. The impairment or overload of these compensatory mechanisms results in compromised cell viability [23]. Cancer cells usually show an elevated basal intracellular level of ROS without harmful consequences due to adaptation mechanisms sustaining tumorigenesis. Further oxidative stress could overcome these mechanisms restoring the vulnerability of cancer cells to ROS-mediated damage, improving therapeutic response [2,24,25].

In this study we investigate the possibility to use ROS-mediated damage to restore CMA inducibility and GBM cell sensitivity to TMZ.

## 2. Materials and Methods

### 2.1. Cell Lines and Reagents

U251 cell line was obtained from Dr. G. Melillo, and T98 cell line was obtained from Dr. V. Vaira and were routinely maintained as adherent cells in RPMI 1640 medium, supplemented with 10% heat-inactivated fetal bovine serum, penicillin and streptomycin (50 IU/mL), and 2 mM glutamine (all Euroclone, Pero, MI, Italy). Cells were maintained in a humidified atmosphere of 5% CO_2_ at 37 °C in normoxia. Cells were plated at 15,000 cells/cm^2^ and after 24 h, in vitro treatments were performed as follow: 100 μM Temozolomide, TMZ, for 24 h (h); 200 μM or 1 mM of Hydrogen Peroxide, H_2_O_2_, for 24 h; 25 μM MitoTEMPO, MitoT, for 1 h of pre-treatment (all Sigma-Aldrich, St. Louis, MO, USA). After treatments samples were analyzed for cell viability through Trypan Blue exclusion test. Glioma cells were transfected for 24 h with 10 nM of *HSC70* or *PHLPP1* siRNA or a scrambled negative control (Eurofins, Italy) in presence of a T-Pro-P-Fect reagent (T-Pro Biotechnology, New Taipei, Taiwan), and then cells were treated with TMZ.

### 2.2. Biochemical Assays

The ROS content after different treatments was tested by using ROS-Glo™ H_2_O_2_ Assay kit (Promega, Milan, Italy). HIF-1α activity was measured on lysates through Luciferase Biochemical assay, using GloMax-Multi Detection System (Promega, Milan, Italy), and normalized for protein content [26]. The cytotoxicity of treatments was tested utilizing Cell Tox™ Green Cytotoxicity Assay kit (Promega, Milan, Italy) and Cell Titer-Glo^®^ Luminescent Cell Viability Assay (Promega). Detection and quantification of Glutathione (GSH) was performed after treatment by the commercially available GSH-Glo™ Glutathione Assay (Promega). Data were expressed as Glutathione concentration. All the assays performed by using commercially available kits were carried out according to the manufacturer’s instructions.

### 2.3. RNA Extraction and Real-Time PCR

RNA was extracted by using a commercially available Illustra RNA spin Mini Isolation Kit (GE Healthcare, Milan, Italy) in accordance with the manufacturer’s instructions. Total RNA was reverse-transcribed to cDNA by using a High-Capacity cDNA Reverse Transcription Kit (Applied Biosystems, Monza, Italy). The real-time PCRs were performed in triplicate for each data point by using the Sybr Green technique; the oligonucleotides used are shown in Table 1. Target mRNA content changes in relation to the *β-ACTIN* housekeeping gene were determined using the ΔΔCt Method (and represented as FOI, fold of induction, compared to control level).

### 2.4. Wounding Assay

For the wound healing assay, at the end of the treatment, a wound was created by manually scraping the confluent glioma monolayer with a p200 pipette tip. Images at time zero (t = 0 h) were acquired to record the initial area of the wounds, and the recovery of the wounded monolayers due to cell migration toward the free area was evaluated at 24 h (t = 24 h). The area of wound was quantified by Java’s Image J software (http://rsb.info.nih.gov) and the migration of cells toward the wounds was expressed as percentage of wound closure: % of wound closure = [(A**_(t_**
_= **0 h)**_−A**_(t = 24h)_**)/A**_(t = 0 h)_**] × 100, where, A**_(t = 0 h)_** is the area of wound measured immediately after scratching, and A**_(t=24h)_** is the area of wound measured 24 h after scratching.

### 2.5. Protein Studies

Samples were prepared in Novex Bolt LDS sample buffer and Novex Bolt Reducing Agent and were boiled for 3 min. Lowry method was used for protein quantification. Proteins (10–30 μg) were loaded in Precast Bolt^®^ Bis-Tris Plus Gels 4–12% and run for 30 min at 200 V in Novex Bolt 1X MES/SDS Running buffer. Transfer to nitrocellulose membrane was performed by a Trans-Blot^®^ Turbo™ system (BioRad, Hercules, CA, USA). Membranes were blocked in Odyssey^®^ Blocking Buffer (LICOR, Lincoln, NE, USA) and incubated with secondary fluorescent antibodies (IRDye^®^, LICOR, Lincoln, NE, USA). Protein bands were visualized by an Odyssey Fc device, model 2800 (LICOR Biosciences). Bands intensity was quantified by Image Studio Lite Ver 5.2 software. Subcellular fractions of cells were obtained using Protein Fractionation Kit (ThermoScientific, Waltham, MA, USA) [27], according to manufacturer’s instruction. Protein signals were normalized for the respective fraction marker: TUBULIN for cytosol and LAMP1 for lysosomes.

Samples were probed with the following antibodies: LAMP2A (51-2200 1:1000, Invitrogen, Carlsbad, CA, USA), HSC70 (MA1-26078 1:000, Invitrogen), GAPDH (sc4772 1:1200, Santa Cruz, Santa Cruz, CA, USA), NDUFB8 (ab110242 1:1000, Abcam, Cambridge, UK), SDHB (ab14714 1:1000, Abcam), UQCRC2 (ab14745 1:1000, Abcam), MTCO2 (ab110258 1:1000, Abcam), ATP5A (ab14748 1:1000, Abcam), SDHA (459200 1:10000, Invitrogen), COXIV (A21347 1:2000, Invitrogen), SOD1 and CATALASE (Abcam, ab179843 1:500), LAMP1 (ab25630 1:900, Abcam), ACTIN (A2066 1:200, Sigma-Aldrich, St. Louis, MO, USA), α-TUBULIN (3873S 1:800, Cell signaling, Danvers, MA, USA).

### 2.6. Mitochondrial Complex Activity Studies

Protein extraction was performed by sonication (50 W for 10 s, 3 times) after suspending cell pellets in the proper buffer (pH 7.2). Lysates were centrifuged at 750× *g* for 10 min and supernatant was recovered. Lowry method was used for protein quantification. A Lambda 2 spectrophotometer (Perkin Elmer, Waltham, MA, USA) was used to assess enzymatic activities. Analyses were performed at specific wavelengths for each enzymatic activity after preparing proper solutions as previously described [27] with minor changes. Experiments were performed at 30 °C. Analyses were performed through the Perkin Elmer software. Measurements were normalized for the activity level of citrate synthase, a stable matrix mitochondrial enzyme; this latter step was performed in order to normalize respiratory chain activity for mitochondrial mass.

### 2.7. Statistical Analyses

The in vitro experiments were repeated at least three times and led to reproducible results. The data are presented as the mean values ± SD of the independent experiments and were statistically analyzed using a t-test or one- or two-way analysis of variance, followed by Dunnett’s or Bonferroni’s multiple comparison and Prism 4 software (GraphPad Software Inc., San Diego, CA, USA).

## 3. Results

### 3.1. Mitochondrial ROS are Crucial for TMZ Responsiveness in U251 Cells

Aiming to characterize ROS involvement in TMZ-sensitivity, first we measured ROS levels in TMZ-sensitive (U251) and TMZ-resistant (T98) GBM cell lines before and after exposure to TMZ. ROS basal levels were 8-fold higher in T98 compared to U251 cells. After 24 h of treatment, TMZ induced a significant increase in ROS levels in U251 sensitive cells but not in T98 resistant cells (Figure 1A).

We replicated the experiment by treating cells with the selective mitochondrial ROS scavenger Mitotempo (MitoT). MitoT was able to reduce ROS levels in both cell lines, confirming their mitochondrial origin, however, residual ROS levels were still higher in TMZ-resistant cells (Figure 1B,C). Interestingly, the inhibition of mitochondrial ROS release by MitoT impaired the cytotoxic effect of TMZ in U251 cells, as assessed by the count of viable cells (Figure 1D) and the evaluation of cellular toxicity (Figure 1E). Global U251 cellular viability was unaffected by MitoT in absence of TMZ. At molecular level, MitoT engaged a pro-survival response in TMZ-treated U251 cells and counteracted the pro-apoptotic gene expression signature induced by TMZ (Figure 1F).

A different behavior was observed in T98 cells: TMZ-resistance was unchanged after TMZ treatment and/or MitoT. Indeed, MitoT was ineffective in modifying cellular viability and toxicity in TMZ-treated T98 cells (Figure 1G,H). Similarly, the “resistant-like” gene expression signature observed in T98 cells after TMZ treatment was preserved after the combined administration of MitoT (Figure 1I).

The functional consequences of the combined TMZ and MitoT treatment were also investigated by monitoring GBM cell motility. The scratch test, performed in U251 TMZ-sensitive cells, suggested that MitoT favored the scratch closure, annulling the anti-proliferative effect of TMZ in these cells (Figure 1L). This conclusion was supported by gene expression profile analysis performed on EMT genes: the reduction of *SLUG* and the increase of *E-CADHERIN* (*E-CAD*) expression in TMZ-treated U251 sensitive cells was partially or almost totally counteracted by MitoT when administered alone or in combination with TMZ, respectively (Figure 1M). Once again, this effect was restricted to sensitive U251 cells (Figure 1N,O).

### 3.2. Involvement of Chaperone Mediated Autophagy in GBM Responsiveness to TMZ

We have previously associated the activation of the CMA and HIF-1α activity reduction with GBM cell responsiveness to TMZ [6,7]. These findings prompted to investigate how MitoT could affect CMA, an intracellular pathway engaged by ROS release [28] and known to modulate HIF-1α-activity [7].

To extend the evaluation of consequences of ROS scavenging on CMA activity and TMZ-responsiveness, we assessed CMA activity before and after MitoT delivery in our experimental conditions by checking the expression levels of three genes involved in the positive regulation of CMA: lysosome-associated membrane protein (*LAMP2A*), heat shock cognate 70 kDa protein (*HSC70*) and pleckstrin homology domain and leucine rich repeat protein phosphatase (*PHLPP1*). Transcript levels were found higher in TMZ-treated versus untreated U251 cells supporting the activation of CMA program in relation to TMZ-responsiveness. This effect was not observed with MitoT co-administration. Notably, CMA-related gene expression profile was significantly reduced by MitoT itself (Figure 2A).

In T98 resistant cells, with or without TMZ, gene expression levels were unaltered by MitoT with the only exception of *LAMP2A* expression, which was found decreased in MitoT-treated cells (Figure 2B).

As regards the reduction of HIF-1α activity in relation to TMZ-responsiveness in sensitive cells [6,7], MitoT prevented this effect in TMZ-treated U251 sensitive cells. Conversely, MitoT produced no modification in cell responsiveness when administered to T98 resistant cells (Figure 2C,D). Quantitative RT-PCR studies, evaluating the expression levels of *HIF-1α* gene (*HIF-1α*) and its established target gene vascular endothelial growth factor (*VEGF*), supported this conclusion (Figure 2E,F), even showing an increase in *HIF-1α* and *VEGF* transcripts in MitoT and TMZ-containing treatments in U251 cells.

### 3.3. Chaperone Mediated Autophagy Activity is Dependent Upon Different Proteins

To check LAMP2A and HSC-70 intracellular localization, we performed cell fractionation in U251 and T98 cells with or without TMZ (Figure 3A,B; Appendix A).

After the normalization for the relative fraction markers, we observed a statistically significant increase in lysosomal recruitment of the CMA receptor LAMP2A and an increase in cytosolic-to-lysosomal translocation of the CMA chaperone HSC70 in TMZ-treated U251 cells (*p* = 0.065) but not in TMZ-treated T98 cells (compared to untreated control level). The lysosomal level of the CMA substrate Glyceraldehyde 3-phosphate dehydrogenase (GAPDH) was also increased in TMZ-treated U251 cells (*p* = 0,059). Overall, these findings support the selective engagement of CMA pathway in TMZ-sensitive cells.

We have previously demonstrated that blocking CMA by *LAMP2A* silencing provoked TMZ-resistance in U251 cells [7]. To expand these observations to other key players of CMA, we evaluated TMZ-responsiveness after transient downregulation of *HSC70* and *PHLPP1* in both TMZ sensitive and resistant cells. In the sensitive U251 cells, siRNA against *HSC70* or *PHLPP1* abolished TMZ-dependent cell death and HIF-1α activity reduction (Figure 3C,D). TMZ-responsiveness and HIF-1α activity in T98 cells was not changed under the same experimental conditions (Figure 3E,F). Transcriptional changes underlining the acquisition of TMZ-resistance were observed in TMZ-treated U251 cells after *HSC70* or *PHLPP1* silencing (Figure 3G), paralleling previous achievements in *LAMP2A*-silenced cells [7]. T98 cells were not affected by CMA downregulation and maintained their resistant-like behavior for CMA-related (Figure 3H) and *HIF-1α*/*VEGF* gene expression (Appendix A).

The investigation of EMT-related gene expression profile showed that, in U251 cells, *HSC70* and *PHLPP1* silencing *per se* did not produce any change in *SNAIL*, *SLUG* and *E-CAD* expression. However, TMZ treatment in silenced cells induced a mesenchymal-like expression profile, with an increase in *SNAIL* and *SLUG* expression and a downregulation of *E-CAD* mRNA level, typical of TMZ-resistant cells (Figure 3I). T98 cells did not show any difference in EMT modulation after TMZ treatment even after *HSC70* and *PHLPP1* silencing (Figure 3L).

### 3.4. Deregulation of Redox-Homeostasis is Involved in GBM Responsiveness to TMZ

To better understand the mechanisms involved in cytoplasmic ROS increase after TMZ treatment in sensitive cells, mitochondrial and cytosolic detox mechanisms were assessed.

Since ROS production and thus redox-homeostasis is dependent by the induction of the respiratory chain, we measured the activities of mitochondrial respiratory chain complexes I, II, I+III and IV in U251 and T98 cells at basal levels and after 24 h treatment with TMZ. We observed that basal activities of complexes I, III and IV were consistently higher in T98 compared to U251 cells and this difference was maintained after treatment (Figure 4A,B).

No differences were observed in citrate synthase levels, ruling out the chance of altered global mitochondrial content (data not shown). A slight reduction of Complex I activity was observed in TMZ-treated U251 cells after 24 h (18%). In T98 cells treatment with TMZ did not produce any reduction of complex activities (Figure 4B).

To check whether these differences are the consequence of altered stability of mitochondrial respiratory chain complexes, we performed SDS-PAGE analysis of representative oxidative phosphorylation system (OXPHOS) subunits in the same samples used for biochemical studies. Despite the detection of heterogeneous signals, we failed to disclose any difference between U251 and T98 cells in basal conditions as well as after TMZ treatment (Figure 4C–E). These findings suggest that increased OXPHOS activities observed in TMZ-resistant cells and the decrease observed after TMZ delivery might originate from deregulated OXPHOS homeostasis and/or altered ROS management while steady state levels of mitochondrial proteins are conserved.

To assess the role of the redox-homeostasis in TMZ-responsiveness, we assessed also the gene expression profile of the main modulators of the detox machinery. In detail we observed that, despite the different basal levels of *SOD* and *CATALASE* expression (Appendix A), in U251 TMZ-responsive cells glutathione peroxidase (*GPX*), glutathione synthetase (*GSS*), *SOD-2* and *CATALASE*, were downregulated by TMZ treatment (Figure 4F). On the contrary, in T98 cells, with the exception of *SOD-2* and *GPX*, all other genes resulted up-regulated by the treatment (Figure 4F). Moreover, GSH amount was analyzed in both cell lines, showing, in addition to the high levels measured in T98 TMZ-resistant cells, that TMZ treatment was able to reduce the GSH concentration only in U251 TMZ-sensitive cells (Figure 4G). Diminished residual levels of CATALASE and SOD-1 after TMZ treatment were also observed at protein level in U251, but not in T98 cells where SOD-1 was even statistically increased after 48h treatment with TMZ (Figure 4H,I).

### 3.5. Chemical-Induced Oxidative Stress Weakens the Resistance of T98 Cells to TMZ

Data so far presented suggest that mitochondrial ROS scavenging drives the transition from TMZ-sensitive to TMZ-resistant phenotype in U251 cells. In the last set of experiments, we aimed to verify whether chemically-induced ROS build-up might influence TMZ-responsiveness in U251 and, more importantly, in T98 cells. For this purpose, CMA-mediated GBM cytotoxicity was assessed in TMZ-sensitive and -resistant cells after treatment with both TMZ and H_2_O_2_. Oxidative stress promoted by low concentrations of H_2_O_2_ (200 μM, “mild” dose) was ineffective in reducing U251 viability while H_2_O_2_ and TMZ co-treatment displayed the same effect observed for single TMZ delivery (Figure 5A).

However, an increase of H_2_O_2_ dose (1 mM, “high” dose) was sufficient to mimic the effect of single TMZ treatment although, even in this case, we did not observe synergy between H_2_O_2_ and TMZ (Figure 5A). The induction of apoptotic gene expression pattern fitted with viability data (Figure 5B). In T98 cells, “mild” oxidative stress failed to perturb cellular survival. On the other hand, the “high” H_2_O_2_ dose produced a significant decrease in cell viability, both alone and in combination with TMZ (Figure 5C). Gene expression studies showed that only the combination of the “high” H_2_O_2_ dose and TMZ was able to redirect the molecular profile towards apoptosis (Figure 5D).

The cytotoxic effect induced by the “high” H_2_O_2_ dose in U251 cells was matched by increased expression of *LAMP2A* and *HSC70* and reduced expression of *HIF-1α* and *VEGF*. Upregulation of all CMA-related genes was obtained only after concurrent TMZ treatment (Figure 5E,F). The same results were described for TMZ-resistant T98 cells (Figure 5G,H), where only the “high” H_2_O_2_ dose produced an upregulation of all CMA genes. These data confirm that induced oxidative stress exerts a cytotoxic effect through CMA activation. Importantly, oxidative stress seems to overcome TMZ-resistance in T98 cells.

Mild oxidative stress was not able to induce any delay in wound closure in absence of TMZ in U251 cells (Figure 5I). However, the “high” H_2_O_2_ concentration promoted the acquisition of an epithelial expression profile overlapping the one obtained after TMZ treatment (Figure 5L). In T98 cells, the efficacy of the combined treatment with the “high” H_2_O_2_ dose and TMZ was also demonstrated by scratch test (Figure 5M). Once again, at the molecular level, only the combined use of TMZ and H_2_O_2_ allowed to achieve a sensitive-like profile as demonstrated by the upregulation of *E-CAD* and the downregulation of *SNAIL* and *SLUG* (Figure 5N).

## 4. Discussion

Results shown herein give emphasis to the importance of ROS in inducing CMA and its fundamental role in determining responsiveness to TMZ. CMA has been described as a sensor of the oxidative stress, being involved in the removal of proteins altered by ROS intracellular activity and concurring in cell homeostasis regulation by selectively degrading specific proteins. It is known that oxidative stress allows the upregulation of CMA key players such as LAMP2A and HSC70 [28].

Despite their oxidizing action, ROS levels are important to maintain cellular homeostasis, also mediating some key transduction pathways [29,30] and promoting apoptotic switch [3] and CMA activation [28], also in hypoxic microenvironment [30].

Here, we described that different ROS levels detected in sensitive and resistant cells after treatment might play an essential role in cell responsivity to TMZ, and in modulating CMA activity.

Starting from ROS analysis, U251 TMZ-responsive cells are characterized by a lower basal level of cytoplasmic ROS compared to T98 TMZ-resistant ones, and only in sensitive cells, TMZ can induce an increase of cytoplasmic ROS. TMZ-induced ROS are fundamental for GBM cell responsiveness driving CMA activation, the expression of pro-apoptotic genes and the re-programming of an epithelial-like expression pattern. On the other hand, the lack in transitory increase of cytoplasmic ROS after TMZ treatment in T98 resistant cell line, avoid CMA activation, pro-apoptotic and pro-epithelial gene expression, confirming the crucial role played by the increase of ROS level in leading the onset of the cytotoxic effect driven by CMA. In this work, we showed that ROS increase is due to their mitochondrial release. Indeed, the use of MitoT, a mitochondrial ROS scavenger, highlighted that mitochondria play a fundamental role in response to TMZ. In fact, the combined use of TMZ with MitoT in U251 sensitive cells, impairs the cytotoxic effect mediated by TMZ treatment by inhibiting mitochondrial ROS release. Future studies will be aimed at elucidating the mechanisms underlying this phenomenon since mitochondria not only are the main ROS producer within the cell, but they are also key regulator of oxidative metabolism, involved also in the apoptotic switch [31].

Treatment with MitoT in U251 cells not only prevented TMZ-dependent decrease in cell viability, the induction of cytotoxicity, the expression of a pro-apoptotic and pro-epithelial gene expression pattern and the decrease in cell motility, but the abrogation of mitochondrial ROS reverted also the TMZ induced pattern of expression of CMA-related genes (*LAMP2A*, *HSC70*). The same effect was observed also for *PHLPP1* expression, which was altered even by MitoT treatment itself. PHLPP1 has been proposed as a negative modulator of tumorigenesis, being associated to the promotion of apoptosis [32] and, in U251 cell line, it was demonstrated to be involved in the suppression of tumor malignancy and in the modulation of inflammatory cytokines [33]. PHLPP1 activity is of great importance also for its role in the positive regulation of CMA activity and in the AKT dephosphorylation [34]. Results described herein propose an important role for PHLPP-1, whose activity could be directly regulated by cytoplasmic ROS levels and switching on CMA.

Supporting the molecular data, CMA activity and inhibition is always in line with HIF-1α expression and activity, and *VEGF* transcript level. HIF-1α is in fact a CMA target and *VEGF* is one of its main direct target genes. For this reason, here we reaffirm the use of HIF-1α activity and *VEGF* expression as biomarkers for CMA activity assessment.

Classically speaking, HIF-1α activity is strictly regulated by oxygen availability whose decrease is able to reduce HIF-1α proteasome degradation by inhibiting Prolyl Hydroxylase (PHD) activity, and consequently, is able to induce an increase of the activity of this important transcription factor. However, many other mechanisms have been described to be able to modulate HIF-1α activity independently from oxygen availability. This is the reason why to clarify the direct relation between GBM responsiveness to TMZ, CMA and HIF-1α activity, we performed all the experiments in normoxia, to prevent a disguise of modulations due to hypoxia. We have previously analyzed the relation between GBM responsiveness to TMZ and hypoxia, reporting that low oxygen condition is able to increase resistance even in previously sensitive cells. Further analyses to elucidate the influence of an increase in HIF-1α activity due to different causes including hypoxia upon resistance and in relation to CMA activity will be carried out as a future development of the project. Major aims will include trying to identify the molecular mechanisms mainly involved, but nowadays, they are beyond the aims of this paper. Indeed, independently from oxygen availability, the regulation played by CMA on HIF-1α activity is a fundamental issue to be investigated in relation to the modulation of cell metabolism and of several processes involved in proliferation, stemness and invasiveness, as we have already demonstrated [7].

To clarify molecular results, it is important to note that CMA is a complex mechanism characterized by the synergic functions of different proteins. The whole CMA mechanism can be influenced at three different levels. A first control level is related to the expression and function of carrier proteins that bind a KFERQ-like motif in specific proteins (HSC70, STUB/CHIP and related proteins) and drive them to the lysosome. The second control level includes the expression and activity of the transporter LAMP2A, whose function is determined by post-translational modifications modulating its multimerization and function. Third level is the real check-point driving activation of CMA activity and is related to the activity of specific modulators such as PHLPP1 and the expression of proteins acting as bridge between CMA players and the lysosome (RAC-1).

Now, it is easy to understand the complexity in CMA activity regulation and our results demonstrate the crucial role of each component of the machinery. As a matter of fact, the silencing of just one player results in the abrogation of the whole mechanism. We had already demonstrated that *LAMP2A* silencing is sufficient to block CMA activity, driving U251 cells toward a resistant profile [7]. Here we show that even the silencing of other CMA players such as *HSC70* or *PHLPP1* is sufficient to block CMA activation reverting TMZ-responsiveness in U251 cells to a resistant phenotype. What is most interesting is that while *HSC70* silencing impaired CMA activity without influencing other CMA-related genes, *PHLPP1* silencing, albeit inducing the same functional effects upon CMA activity, was also able to negatively modulate the other CMA related genes (*LAMP2A* and *HSC70*), after treatment with TMZ, exactly reproducing TMZ effects occurring in naturally resistant cells. All these data suggested an essential role for CMA and in particular for PHLPP1-regulated pathways in relation to TMZ responsiveness. In T98 cells, the silencing of CMA-related genes did not produce differences in responsiveness to TMZ but at the molecular level, *PHLPP1* silencing statistically induced the expression of *HSC70*. This phenomenon could be due to the modulation of a phosphorylation-dependent pathway involved in *HSC70* modulation, but the exact explanation of this mechanism is beyond the aim of this study and will be investigated in the future.

Another aim of this study was the elucidation of the mechanisms involved in the increase of cytoplasmic ROS after treatment with TMZ. ROS are mainly produced by mitochondria respiratory chain, but cells have developed several mechanisms to avoid their detrimental effect within the cells. Our results demonstrated that basal differences in ROS levels between sensitive and resistant cells are mainly due to a different activity of the respiratory chain, but that this mechanism is not involved in the transitory increase of ROS level measured after TMZ treatment in sensitive U251 cells.

Similarly, the stability of respiratory chain complexes is not modified by treatment and cannot account for ROS release after treatment. Resistance to therapy has recently been associated to the activation of anti-oxidant cell species, also in GBM [35]. For instance, the activation of GR [35], CATALASE [36] and SOD-2 [37] was higher in TMZ-resistant cells, providing evidences of their involvement in mediating resistance. Results described herein confirm these data showing differential expression and activity of genes involved in detoxification from ROS (*GSH*, *GSS*, *GPX*, *CATALASE*, *SOD-2*).

Several therapeutic approaches aimed at reducing cell detoxifying agents in tumor cells have already been described [38,39], as well as several treatments inducing an increase in oxidative stress (e.g., the Stupp protocol itself [1]). Here we have demonstrated that an exogenous increase in intracellular ROS level induces the same mechanisms activated by TMZ-dependent ROS release in sensitive cells and is able to overcome detox systems in resistant ones. In detail, 1 mM H_2_O_2_ treatment in U251 cells was able to induce a pro-apoptotic pattern of expression, to activate CMA and to determine an epithelial-like expression profile, similarly to what happens after treatment with TMZ. On the other hand, in T98 TMZ-resistant cells, even if 1 mM H_2_O_2_ treatment produced a reduction in cell viability and CMA activation, a completely responsive-like gene expression pattern for both apoptosis, CMA, and EMT was obtained only after concurrent TMZ treatment. These results highlight the importance of the coexistence of TMZ molecular effects with an increase in intracellular oxidative stress and the fundamental role played by CMA in mediating its cytotoxicity.

## 5. Conclusions

This work demonstrates that CMA activity is induced by a transitory increase in intracellular ROS level, and that GBM cell sensitivity to TMZ is strictly related to this ROS temporary increment. Sensitive and resistant cells show differences in detox system that could explain the phenotypic differences between TMZ-sensitive and -resistant cells. Starting from these considerations, two main conclusions can be drawn. First, TMZ drives the burst of mitochondrial ROS in sensitive cells leading to cell death and the presence of ROS is crucial for TMZ-responsiveness since ROS abrogation blocks its effects. Second, mitochondrial ROS release drives CMA activation and this phenomenon is essential for inducing cell toxicity by treatment. Overall these findings provide evidences for improving the design of innovative GBM therapies aimed at overcoming resistance by inducing CMA activity.

## Figures and Tables

**Figure 1 cells-08-01315-f001:**
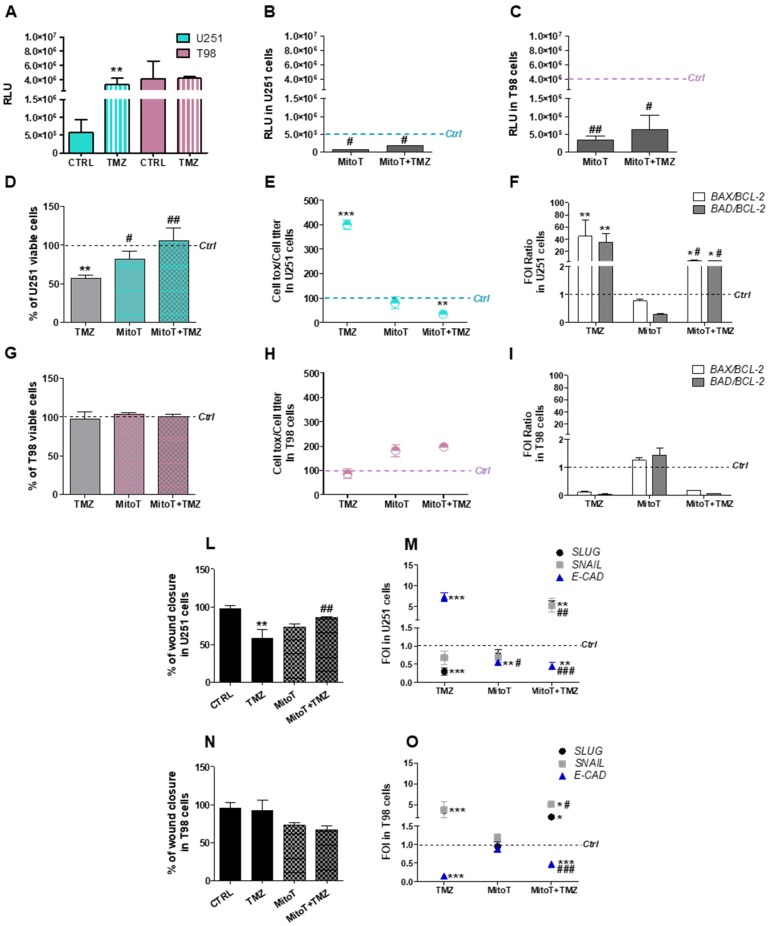
Crucial role of mitochondrial reactive oxygen species (ROS) in Temozolomide-responsiveness in U251 and T98 cells. (**A**) Luminescent assay applied to measure H_2_O_2_ levels in cell culture medium of U251 and T98 cells in untreated cells and after 24 h treatment with 100 µM Temozolomide (TMZ). Data were expressed as relative luminescence units (RLU) obtained by luciferase counts normalized for the amount of proteins quantified by Bradford assay. ** *p* < 0.01 vs. control cells. (**B**) ROS levels measured in U251 and (**C**) T98 cells after 1h of treatment with MitoTempo (MitoT) 25 µM ± TMZ for 24 h. Data were expressed as RLU. # *p* < 0.05, ## *p* < 0.01 vs. TMZ-treated cells. (**D**) Viability of U251 and (**G**) T98 cells, assessed by means of Trypan blue exclusion test, and expressed as the percentage of viable cells after treatment with 100 µM TMZ ± 25 µM MitoT. ** *p* < 0.01 vs. control cells; # *p* < 0.05, ## *p* < 0.01 vs. TMZ-treated cells. (**E**) Cell toxicity in U251 and (**H**) T98 cells analyzed by Cell Tox Green normalized on Cell Titer Glo and expressed as percentage compared to control cells. *** *p* < 0.001 vs. control cells. (**F**) Gene expression analysis for *BAX*, *BAD* and *BCL-2* analyzed by means of Real-time PCR in U251 and (**I**) T98 cells after treatment with 100 µM TMZ ± 25 µM MitoT. Data were normalized to *β-ACTIN*, and the ΔΔCt values were expressed as fold of induction (FOI) of the ratio between treated and control cells and then as the ratio *BAX*/*BCL-2* and *BAD*/*BCL-2*. * *p* < 0.05; ** *p* < 0.01; *** *p* < 0.001 treated vs. control cells. # *p* < 0.05, ## *p* < 0.01, ### *p* < 0.001 vs. TMZ-treated cells. (**L**) Scratch test performed after treatments in U251 and (**N**) in T98 cells. Wound closure percentage compared to controls was analyzed with Image J software. ** *p* < 0.01 treated vs. control cells. ## *p* < 0.01 vs. TMZ-treated cells. (**M**) Gene expression analysis for EMT-related genes (*SLUG*, *SNAIL*, *E-CADHERIN* -*E-CAD*-) analyzed by means of Real-time PCR in U251 and (**O**) in T98 cells. Data were normalized and expressed as mentioned above. ** *p* < 0.01; *** *p* < 0.001 treated vs. control cells. ## *p* < 0.01, ### *p* < 0.001 vs. TMZ-treated cells. Mean values ± SD of three independent experiments.

**Figure 2 cells-08-01315-f002:**
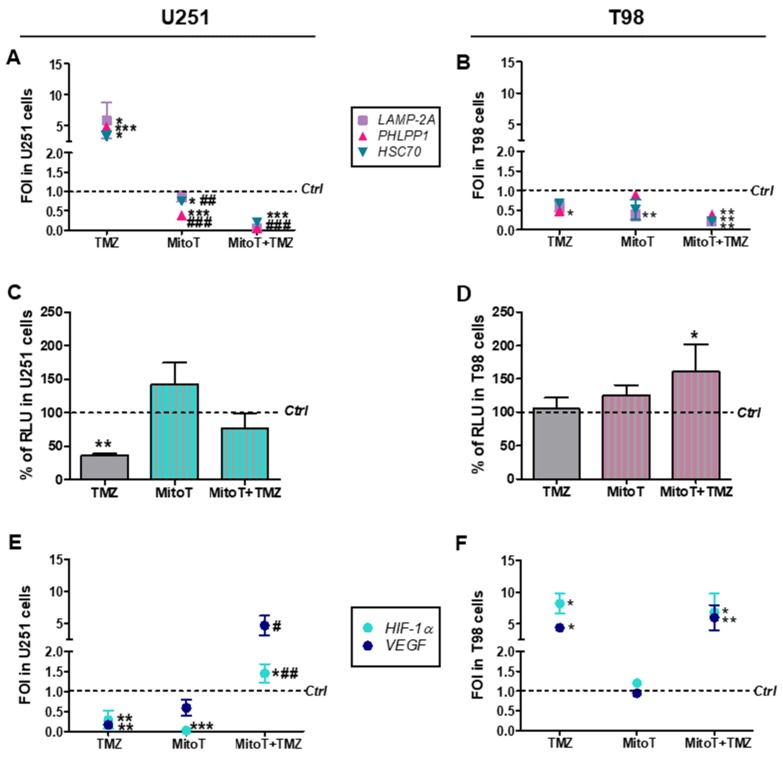
Involvement of chaperone-mediated-autophagy (CMA) after Temozolomide and MitoTempo (MitoT) treatment. (**A**) Gene expression analysis for CMA-related genes (*LAMP2A*, *HSC70*, *PHLPP1*) analyzed by means of Real-time PCR in U251 and (**B**) T98 cells after treatment with 100 µM Temozolomide (TMZ) ± 25 µM MitoT. Data were normalized for *β-ACTIN*, and the ΔΔCt values were expressed as FOI of the ratio between treated and control cells. * *p* < 0.05; ** *p* < 0.01; *** *p* < 0.001 treated vs. control cells. # *p* < 0.05, ## *p* < 0.01, ### *p* < 0.001 vs. TMZ-treated cells. (**C**) Biochemical assay for HIF-1α activity in U251 and (**D**) T98 cell lines. Data were expressed as RLU, obtained normalizing luciferase counts for the amount of proteins quantified by Bradford assay. ** *p* < 0.01 vs. control cells. (**E**) Gene expression analysis for *HIF-1α* and vascular endothelial growth factor (*VEGF*) expression in U251 and (**F**) T98 cells. * *p* < 0.05; ** *p* < 0.01 vs. control cells; # *p* < 0.05, ## *p* < 0.01 vs. TMZ-treated cells. Mean values ± SD of three independent experiments.

**Figure 3 cells-08-01315-f003:**
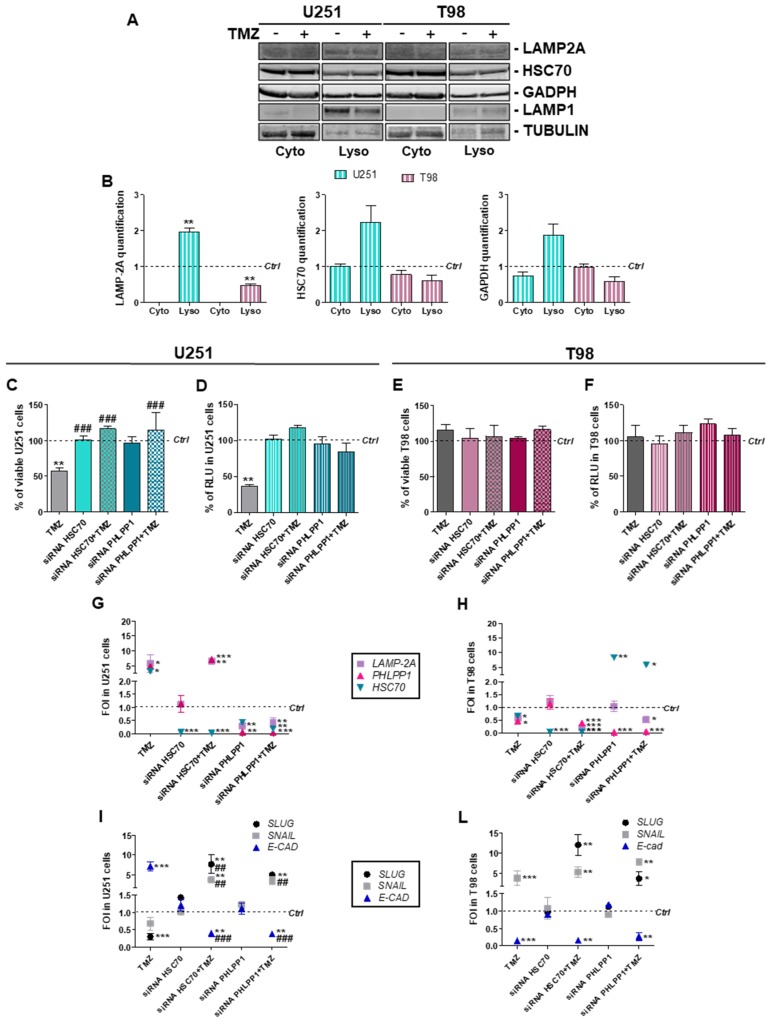
Essential chaperone-mediated-autophagy related gene involvement in Temozolomide response in U251 and T98 cells. (**A**) Western blot analysis of LAMP2A, HSC70 and GAPDH protein levels in subcellular fractions (Cyto = cytosolic fraction; Lyso = Lysosomal fraction) obtained from untreated (−) and Temozolomide (TMZ) treated (+) U251 and T98 cell lines. (**B**) Protein signals were normalized for the respective fraction marker: TUBULIN (cytosol) and LAMP1 (lysosomes). Histograms show the distribution of target proteins in the fractions compared to untreated control levels. ** *p* < 0.01 treated vs. control cells. Lysosomal content of the three considered proteins are significantly different in U251 compared to T98 cells (LAMP2A *p* = 0.003; HSC70 *p* = 0.04; Glyceraldehyde 3-phosphate dehydrogenase (GAPDH) *p* = 0.03). (**C**) Viability analysis and (**D**) assessment of HIF-1α activity in U251 and in (**E**,**F**) T98 cells after *PHLPP1* or *HSC70* silencing ± 100 µM TMZ treatment. Data were expressed as percentage of viable cells and as RLU, as previously described. ** *p* < 0.01 treated vs. control cells; ### *p* < 0.001 vs. TMZ-treated cells. (**G**) Gene expression profile for CMA-related genes (*LAMP2A*, *HSC70*, *PHLPP1*) in U251 and (**H**) T98 cells. (**I**) Gene expression profile for EMT-related genes (*SLUG*, *SNAIL*, *E-CAD*) in U251 and (**L**) T98 cells. Gene expression was analyzed by means of Real-time PCR. All data were normalized for *β-ACTIN*, and the ΔΔCt values were expressed as FOI of the ratio between treated and control cells. * *p* < 0.05; ** *p* < 0.01; *** *p* < 0.001 treated vs. control cells; ## *p* < 0.01, ### *p* < 0.001, vs. TMZ-treated cells. Mean values ± SD of three independent experiments.

**Figure 4 cells-08-01315-f004:**
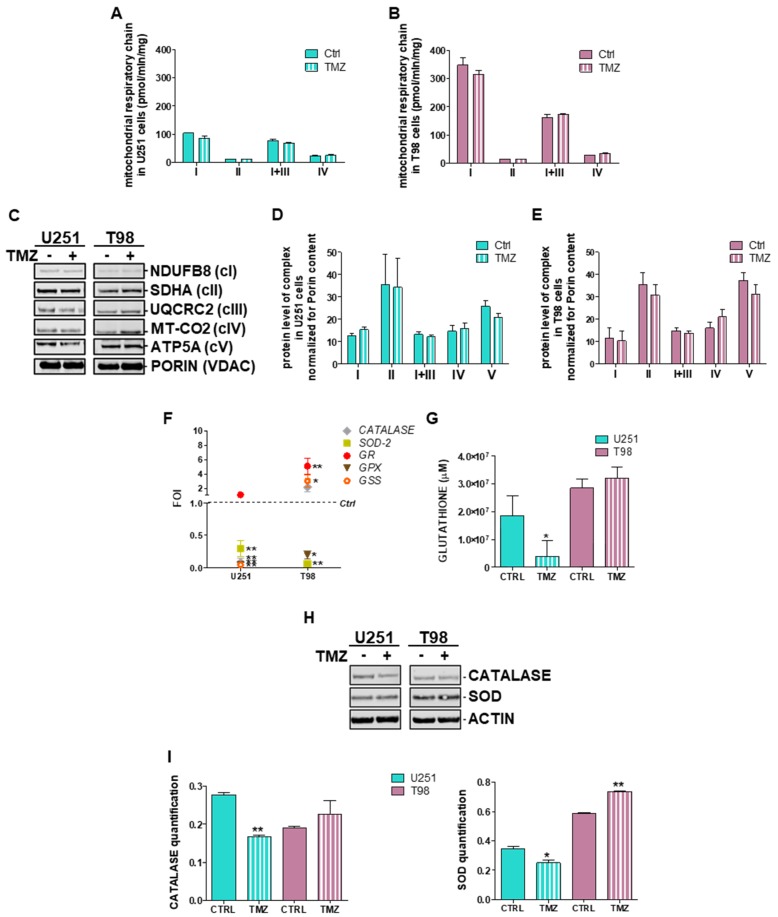
Deregulation of redox-homeostasis after Temozolomide treatment in U251 and T98 cells. (**A**) Spectrophotometric analysis of mitochondrial respiratory chain complex activities measured before and after 24 h of treatment with Temozolomide (TMZ) in U251 and (**B**) T98 cells. Values are expressed as mean values of complex I, II, I+III and IV normalized for citrate synthase activities (pmol/min/mg of proteins). (**C**) Descriptive image of Western Blot analysis of representative subunits of the mitochondrial respiratory chain assessed before and after 24 h of treatment with TMZ and their quantification in (**D**) U251 and (**E**) T98 cells. Values are expressed as mean values of protein content (arbitrary units) normalized for the signal of PORIN (VDAC). (**F**) Gene expression profile for detox enzymes (*SOD-2*, *CATALASE*, *GLUTATHIONE PEROXIDASE* -*GPX-*, *GLUTATHIONE SYNTHETASE* -*GPS-*, and *REDUCTASE* -*GR-*) after treatment with 100 µM TMZ in U251 and T98 cells. Data were normalized for *β-ACTIN*, and the ΔΔCt values were expressed as FOI of the ratio between treated and control cells. * *p* < 0.05; ** *p* < 0.01 vs. control cells. (**G**) Glutathione concentration assessed after TMZ treatment in both cells by means of a commercially available kit. Data were expressed as glutathione concentration (μM). * *p* < 0.05 treated vs. control cells. (**H**) Western blot analysis of CATALASE and SOD in protein lysates from untreated (−) and TMZ-treated (+) U251 and T98 cell lines at 24 h of treatment, and (**i**) their quantification. Protein signals were normalized to ACTIN levels. * *p* < 0.05; ** *p* < 0.01 treated vs. control. Mean values ± SD of three independent experiments.

**Figure 5 cells-08-01315-f005:**
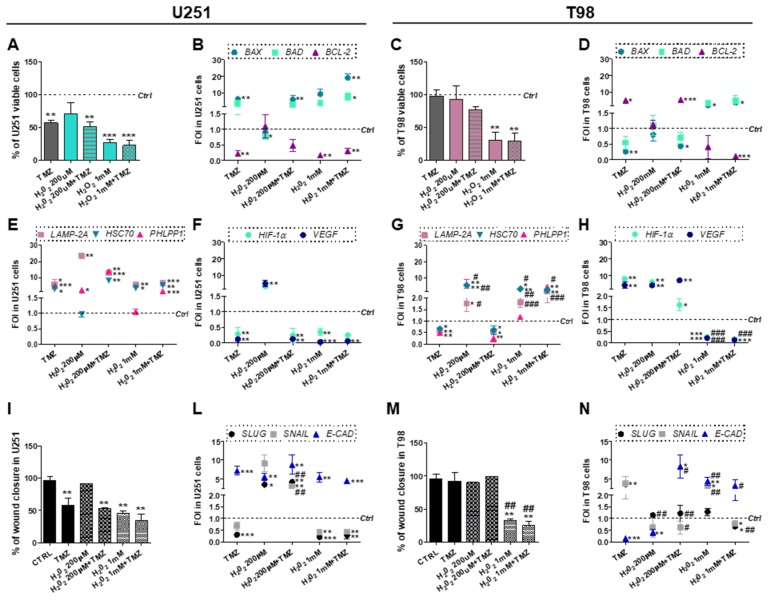
Role of oxidative stress in overcoming Temozolomide resistance. (**A**) Viability analysis and (**B**) gene expression profile for apoptotic-related genes (*BAX*, *BAD* and *BCL-2*) in U251 and (**C**,**D**) in T98 cells. Viability was assessed by means of Trypan blue exclusion test after treatment with 200µM or 1mM H_2_O_2_ ± 100 µM Temozolomide (TMZ). Data of viability were expressed as percentage of viable cells; ** *p* < 0.01; *** *p* < 0.001 treated vs. control cells. (**E**) Gene expression profile for CMA-related genes (*LAMP2A*, *HSC70*, *PHLPP1*), and for (**F**) *HIF-1α* and *VEGF* in U251 and in (**G**,**H**) T98 cell after treatment with 200 µM or 1 mM H_2_O_2_ ± 100 µM TMZ. * *p* < 0.05; ** *p* < 0.01; *** *p* < 0.001 treated vs. control cells. All data of gene expression were normalized for *β-ACTIN*, and the ΔΔCt values were expressed as FOI of the ratio between treated and control cells. * *p* < 0.05; ** *p* < 0.01; *** *p* < 0.001 treated vs. control cells. # *p* < 0.05, ## *p* < 0.01, ### *p* < 0.001 vs. TMZ-treated cells. (**I**) Scratch test and (**L**) gene expression for EMT-related genes (*SLUG*, *SNAIL*, *E-CAD*) performed after treatments in U251 and (**M**,**N**) T98 cells. Wound closure percentage compared to control was analyzed with Image J software. ** *p* < 0.01 treated vs. control cells. ## *p* < 0.01 vs. TMZ-treated cells. Data of gene expression were normalized and expressed as mentioned above. * *p* < 0.05; ** *p* < 0.01; *** *p* < 0.001 treated vs. control cells. # *p* < 0.05, ## *p* < 0.01, ### *p* < 0.001 vs. TMZ-treated cells.

**Table 1 cells-08-01315-t001:** Primer sequences.

Gene	Forward	Reverse
*BAX*	ATGGACGGGTCCGGGGAG	ATCCAGCCCAACAGCCGC
*BAD*	CCCAGAGTTTGAGCCGAGTG	CCCATCCCTTCGTCGTCCT
*BCL-2*	GATTGTGGCCTTCTTTGAG	CAAACTGAGCAGAGTCTTC
*PHLPP1*	CCTACCTTCTCCAGTGCACT	CCAGCAGTTCCAAGTTTCCT
*LAMP2A*	TGCTGGCTACCATGGGGCTG	GCAGCTGCCTGTGGAGTGAGT
*HSC70*	ATTGATCTTGGCACCACCTA	GGGTGCAGGAGGTATGCCTGTGA
*HIF-1* *α*	TGATTGCATCTCCATCTCCTAC	GACTCAAAGCGACAGATAACACG
*VEGF*	CGAGGGCCTGGAGTGTGT	CGCATAATCTGCATGGTGATG
*SNAIL*	GCGAGCTGCAGGACTCTAAT	CCCGCAATGGTCCACAAAAC
*SLUG*	CATGCCTGTCATACCACAAC	GGTGTCAGATGGAGGAGG
*E-CAD*	GATCAAGTCAAGCGTGAGTCG	AGCCTCT CAATGGCGAACAC
*SOD-2*	TTAACGCGCAGATCATGCA	GGTGGCGTTGAGATTGTTCA
*CATALASE*	TAAGACTGACCAGGGCA	CAAACCTTGGTGAGATCGAA
*GR*	AACATCCCAACTGTGGTCTTCAGC	TTGGTAACTGCGTGATACATCGGG
*GPX*	CGCAACGATGTTGCCTGGAACTTT	AGGCTCGATGTCAATGGTCTGGAA
*GSS*	ATGCTGTGCAGATGGACTTCAACC	TGGATGTCAAACAGACGAGCGGTA
*β-ACTIN*	TCAAGATCATTGCTCCTCCTG	CCAGAGGCGTACAGGGATAG

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
