# Peer review of "Intracellular Redox-Balance Involvement in Temozolomide Resistance-Related Molecular Mechanisms in Glioblastoma"

_cells, 2019, doi:10.3390/cells8111315_

Round 1

Reviewer 1 Report

Major revision:

The development of drug resistance, redox balance and HIF-1a-mediated response in cancer cells are very much related with oxygen level in the cellular environment. There is no information provided about oxygen concentration in Methods or Results section, neither possible influence of oxygen availability on the results is covered in discussion. However, this aspect is important enough to be taken into account while designing experimental strategy and at least has to be properly discussed providing indicatory landmarks for the interpretation of the results and future research.

Minor revision:

Some of the methods could be described in more detail. There is no description about how cell fractionation to lysosomes and cytosol for protein studies was performed. While describing cell culture conditions, oxygen concentration is not provided. It is not clear what was cell growth method (g., suspension or adherent cells), dish format, seeding density, how cells were collected for further studies, what were transfection conditions. All experimental steps not covered in the indicated references have to be provided without gaps. The equation for calculation of wound healing efficiency in 2.4. is simple yet looks confusing - it would be more clear if t=0h and t=24h would be subscripted or in brackets. Some of the titles in the 2. Materials and methods section could be more descriptive and related to the results they generate. For example, there are two sections with nearly the same title: 2.2. Biochemical assay and 2.6. Biochemical studies. The abbreviations have to be explained when used for the first time. For example, abbreviation “CMA” is used for the first time in line 33, but explained in line 39. Figure quality has to be improved – font of scale titles, legends and tick labels should be increased, and also the symbols in scatter-plots, so that the information would be clearly readable and not blurred at 100% magnification. The figure titles are incomplete for the figures to standalone. In the title of Figure 2 “Involvement of CMA after TMZ and MitoT treatment”, neither object of treatment, nor that of involvement is indicated, and there are two abbreviations plus a specific name used. In Figure 1 and 3 titles – the type of cells should be added, in the title of Figure 4 – the potential effector of redox homeostasis and the investigated object. Besides incompleteness, the title of Figure 5 has a sense issue, because it means testing the effect of TMZ in TMZ-resistance overcoming. The Figure section indicators should have the same case (either upper or lower) in Figures, legends and text. In Figure 1, description of subsection (m) and (n) is mixed up Section, subsection and figure titles in the study are used in a first lower case format except abbreviation-generating names. However, some words are used first upper case without a clear reason, like “Lines” in section 2.1. title, or “Redox” in Figure 4 title. They have to be changed to the lower case. There are some spelling and grammar issues, like in Lines 94, 143, 145 – “normalize on” or “normalize over” (suggested “normalize for”), or in 2.1. title “reagent” (suggested “reagents”), in line 98 - “expresses ”instead of “expressed”. Thus, the text has to be carefully read and spell-checked.

Author Response

The authors would like to thank the reviewer for its comments. Here, you can fine our answers to each specific issues.

Reviewer #1

1) The development of drug resistance, redox balance and HIF-1a-mediated response in cancer cells are very much related with oxygen level in the cellular environment. There is no information provided about oxygen concentration in Methods or Results section, neither possible influence of oxygen availability on the results is covered in discussion. However, this aspect is important enough to be taken into account while designing experimental strategy and at least has to be properly discussed providing indicatory landmarks for the interpretation of the results and future research.

Author response: The authors are grateful to the Reviewer for the comment. Hypoxia is an important characteristic in glioma, and is crucial for HIF-1α activity up-regulation involved also in tumour progression. However, HIF-1a activity can be modulated by other different pathways independent from oxygen availability.  Main aim of this project has been the clarification of the molecular relation between GBM responsiveness to TMZ, CMA and HIF-1α activity, and we have preferred to perform experiments in normoxia to prevent a maximizing induction of HIF-1a activity due to hypoxia that could mask lower different regulatory mechanisms on HIF-1α activity. To elucidate this concept, we added a more detailed explanation in the Discussion section, lines 411-426, and the extract of added text is reported below.

Classically speaking, HIF-1α activity is strictly regulated by oxygen availability whose decrease is able to reduce HIF-1α proteasome degradation by inhibiting Prolyl Hydroxylase (PHD) activity, and consequently, to induce an increase of the activity of this important transcription factor. However, many other mechanisms have been described to be able to modulate HIF-1α activity independently from oxygen availability. This is the reason why to clarify the direct relation between GBM responsiveness to TMZ, CMA and HIF-1α activity, we performed all the experiments in normoxia, to prevent a disguise of modulations due to hypoxia. We have previously analyzed the relation between GBM responsiveness to TMZ and hypoxia, reporting that low oxygen condition is able to increase resistance even in previously sensitive cells. Further analyses to elucidate the influence of an increase in HIF-1α activity due to different causes, including hypoxia, upon resistance and in relation to CMA activity will be carried out as a future development of the project. Major aims will include trying to identify the molecular mechanisms mainly involved, but nowadays, they are beyond the aims of this paper. Indeed, independently from oxygen availability, the regulation played by CMA on HIF-1α activity is a fundamental issue to be investigated in relation to the modulation of cell metabolism and of several processes involved in proliferation, stemness and invasiveness, as we have already demonstrated[7].

2) Minor revision:

Author response: we thank the Reviewer for all suggestions. Following, we have addressed each specific comments.

Some of the methods could be described in more detail. There is no description about how cell fractionation to lysosomes and cytosol for protein studies was performed.

Author response: Thank you for this comment. We modified the text accordingly.n the text, as regards cell fractionation, we have used the Protein Fractionation Kits (ThermoScientific, Waltham, MA US), according to manufacturer’s instruction and protein signals were normalized for the respective fraction marker: TUBULIN (cytosol) and LAMP1 (lysosomes). Now, we have explained it in the text (lines 130-132).

While describing cell culture conditions, oxygen concentration is not provided. It is not clear what was cell growth method (g., suspension or adherent cells), dish format, seeding density, how cells were collected for further studies, what were transfection conditions. All experimental steps not covered in the indicated references have to be provided without gaps.

Author response: We added the more accurate description of methods. The experiments were carried out with an Oxygen concentration of 20%, considered as normoxia. Cells were growth in adhesion, in petri or different multi well plates, in relation to the different assays (for example, in a 24 well plate for analysis of Hif-1α activity and cell counts, in 96 well for of ROS detection kit, in a 10-cm Petri plate for experiments for protein extracts), always with the same density of 15.000 cells/cm2. Transfection experiments were performed using T-Pro-P-Fect reagent, according to manufacturer’s instructions, for 24 hours, then cells were treated according to the appropriate protocol for different analyses. All these details have been added in the lines 82-92 and 101-102.

The equation for calculation of wound healing efficiency in 2.4. is simple yet looks confusing - it would be more clear if t=0h and t=24h would be subscripted or in brackets.

Author response: Thank you for this advice, we put the terms t=0h and t=24h in bracket and subscripted, to clarify the equation. Now, it is defined as follows: % of wound closure=[(A(t=0h)−A(t=24h))/A(t=0h)]×100 (lines 118-120).

Some of the titles in the 2. Materials and methods section could be more descriptive and related to the results they generate. For example, there are two sections with nearly the same title: 2.2. Biochemical assay and 2.6. Biochemical studies.

Author response: The authors agree with the reviewer, we changed the title of 2.6 section. Instead of 2.6. Biochemical studies, we titled the paragraph as follows:  2.6 Mitochondrial complex activity studies (line 141).

The abbreviations have to be explained when used for the first time. For example, abbreviation “CMA” is used for the first time in line 33, but explained in line 39.

Author response: Thank you, we checked all the abbreviations, and we inserted the explanation the first time that abbreviations appeared. More in detail:

-       CMA was defined in line 30 instead of 33 (and again, definition and abbreviation are both present in the introduction)

-       ROS was defined in lines 30-31 instead of 38 (and again, definition and abbreviation are both present in the introduction)

-       We have introduced the definition of used abbreviations in all the figure legends (such as Temozolomide, TMZ and MitoTempo, MitoT; an example is given in lines 165 and 171 for Figure 1) for a better comprehension of figures, with the aim to become standalone.

-       FOI as fold of induction was added in lines 175

-       E-CAD as E-CADHERIN was added in lines 180 and 199

-       VEGF as vascular endothelial growth factor was added in line 235

-       GAPDH as Glyceraldehyde 3-phosphate dehydrogenase was added in line 245 in Figure 3 and in line 261 in main text

-       OXPHOS as Oxidative Phosphorylation System was added in lines 311-312

Figure quality has to be improved – font of scale titles, legends and tick labels should be increased, and also the symbols in scatter-plots, so that the information would be clearly readable and not blurred at 100% magnification.

Author response: We have modified the font of scale titles, numbers, legends and the tick labels of all figures. Moreover, the symbols in all scatter-plots were increased. In the revised text we included only the revised figures while old figures were deleted.

The figure titles are incomplete for the figures to standalone. In the title of Figure 2 “Involvement of CMA after TMZ and MitoT treatment”, neither object of treatment, nor that of involvement is indicated, and there are two abbreviations plus a specific name used. In Figure 1 and 3 titles – the type of cells should be added, in the title of Figure 4 – the potential effector of redox homeostasis and the investigated object. Besides incompleteness, the title of Figure 5 has a sense issue, because it means testing the effect of TMZ in TMZ-resistance overcoming.

Author response: The agree with the reviewer and we have modified the titles of figures, as reported following:

-       “Figure 1. Crucial role of mitochondrial Reactive Oxygen Species in Temozolomide-responsiveness in U251 and T98 cells” (instead of: Crucial role of mitochondrial ROS in TMZ-responsiveness)

-       “Figure 2. Involvement of chaperone-mediated-autophagy after Temozolomide and MitoTempo treatment” (instead of: Involvement of CMA after TMZ and MitoT treatment)

-       “Figure 3. Essential chaperone-mediated-autophagy related gene involvement in Temozolomide response in U251 and T/98 cells” (instead of: Essential CMA-related gene involvement in TMZ response)

-       “Figure 4. Deregulation of redox-homeostasis after TMZ treatment in U251 and T98 cells” (instead of: Deregulation of Redox-homeostasis)

-       “Figure 5. Role of oxidative stress in overcoming Temozolomide resistance” (instead of: Oxidative stress and TMZ in resistance overcoming).

The Figure section indicators should have the same case (either upper or lower) in Figures, legends and text.

Author response: We have changed the Figure section indicators according to the reviewer’s advice.

In Figure 1, description of subsection (m) and (n) is mixed up.

Author response: We have modified the (M) and (N) order in the legend of Figure 1.

Section, subsection and figure titles in the study are used in a first lower case format except abbreviation-generating names. However, some words are used first upper case without a clear reason, like “Lines” in section 2.1. title, or “Redox” in Figure 4 title. They have to be changed to the lower case.

Author response: We have substituted the first upper case according to the reviewer’s advice as follows:

-       “2.1 Cell lines and reagents” instead of “2.1 Cell Lines and reagent” (line 81)

-       “3.2 Involvement of chaperone mediated autophagy in GBM responsiveness to TMZ” instead of “3.2 Involvement of Chaperone Mediated Autophagy in GBM responsiveness to TMZ” (line 203)

-       “lysosome-associated membrane protein (LAMP2A), heat shock cognate 70 kDa protein (HSC70) and pleckstrin homology domain and leucine rich repeat protein phosphatase (PHLPP1)” instead of “Lysosome-associated membrane protein (LAMP2A), Heat shock cognate 70 kDa protein (HSC70) and Pleckstrin homology domain and leucine rich repeat protein phosphatase (PHLPP1)” (lines 211-213)

-       “3.4 Deregulation of redox-homeostasis is involved in GBM responsiveness to TMZ” instead of “3.4 Deregulation of Redox-homeostasis is involved in GBM responsiveness to TMZ” (line 281)

-       “Gene expression profile for detox enzymes” instead of “Gene expression profile for Detox enzymes” (line 297)

There are some spelling and grammar issues, like in Lines 94, 143, 145 – “normalize on” or “normalize over” (suggested “normalize for”), or in 2.1. title “reagent” (suggested “reagents”), in line 98 - “expresses ”instead of “expressed”. Thus, the text has to be carefully read and spell-checked.

Author response: The authors are sorry for the mistakes. We have checked the main text, we modified text as follows:

-       The word “reagent” in line 81 was substituted with “reagents”

-       “Normalized on”, “normalized over” and “normalized to” were changed in “normalized for” in lines 96-97, 148, 150, 166, 221, 224, 245, 255, 293, 299, 344-345

-       The abbreviation “h” in line 88 was substitute with “hours” or “hour”

-       The verb “expresses” in line 100 was substitute with “expressed”

-       The word “RNAi” in line 268 was substitute with “siRNA”

-       The verb “is” in line 401 was substitute with “was”

Moreover, all text has been checked to uniform the font for what concerns the italic words and the upper and lower case (such as Stupp protocol instead of STUPP protocol in line 474).

A paragraph in the discussion regarding PHPPL1 silencing has been moved from lines 427-434 to lines 399-406, to allow a better reading of the text, having inserted a new part concerning the conditions for HIF-1α activity assessment and its relation to hypoxia/normoxia, for what concerns the experimental setting.

Reviewer 2 Report

The manuscript is very interesting and is well-structured in general. The Authors conducted number of assays to reveal the intracellular redox-balance role in the temozolomide resistance-related molecular mechanisms in glioblastoma. The topic of this study is very interesting. Moreover, in my opinion it could be interesting for a reasonable number of scientists since malignant glioblastoma is characterized by a dismal prognosis manly due to the resistance to conventional therapies. I recommend publication of this in its present form.

Author Response

Reviewer #2

The manuscript is very interesting and is well-structured in general. The Authors conducted number of assays to reveal the intracellular redox-balance role in the temozolomide resistance-related molecular mechanisms in glioblastoma. The topic of this study is very interesting. Moreover, in my opinion it could be interesting for a reasonable number of scientists since malignant glioblastoma is characterized by a dismal prognosis manly due to the resistance to conventional therapies. I recommend publication of this in its present form.

Author Response: We thank the Reviewer for the encouraging comments. The authors hope that the reported findings could improve the knowledge about CMA role in GBM, and that will allow to help in the identification of procedures to overcome resistance of glioma to standard therapy with Temozolomide.